Dominance hierarchies, diversity and species richness of vascular plants in an alpine meadow: contrasting short and medium term responses to simulated global change

Alatalo Juha M. 1 juha.alatalo@ebc.uu.se
Little Chelsea J. 1
Jägerbrand Annika K. 2
Molau Ulf 3
1 Department of Ecology and Genetics, Uppsala University , Visby , Sweden
2 VTI, Swedish National Road and Transport Research Institute , Stockholm , Sweden
3 Department of Biological and Environmental Sciences, University of Gothenburg , Gothenburg , Sweden
Buckley Hannah
Electronic publication date: 2014 May 22
Publication date: 2014
Volume: 2
Electronic Location ID: e406
Received 2014 Mar 17; Accepted 2014 May 8
Copyright: © 2014 Alatalo et al.
Copyright year: 2014
Copyright holder: Alatalo et al.
License: This is an open access article distributed under the terms of the Creative Commons Attribution License, which permits unrestricted use, distribution, reproduction and adaptation in any medium and for any purpose provided that it is properly attributed. For attribution, the original author(s), title, publication source (PeerJ) and either DOI or URL of the article must be cited.
License URL: https://creativecommons.org/licenses/by/4.0/

Keywords: Alpine tundra, Climate change, Plant community diversity, Meadow, Functional groups, Nutrient addition, Species richness, Warming, Global change, Arctic

Funding: Oscar och LiIli Lamms Minne This study was financially supported by Oscar och LiIli Lamms Minne to JMA. The funder had no role in study design, data collection and analysis, decision to publish, or preparation of the manuscript.

==============================
We studied the impact of simulated global change on a high alpine meadow plant community. Specifically, we examined whether short-term (5 years) responses are good predictors for medium-term (7 years) changes in the system by applying a factorial warming and nutrient manipulation to 20 plots in Latnjajaure, subarctic Sweden. Seven years of experimental warming and nutrient enhancement caused dramatic shifts in dominance hierarchies in response to the nutrient and the combined warming and nutrient enhancement treatments. Dominance hierarchies in the meadow moved from a community being dominated by cushion plants, deciduous, and evergreen shrubs to a community being dominated by grasses, sedges, and forbs. Short-term responses were shown to be inconsistent in their ability to predict medium-term responses for most functional groups, however, grasses showed a consistent and very substantial increase in response to nutrient addition over the seven years. The non-linear responses over time point out the importance of longer-term studies with repeated measurements to be able to better predict future changes. Forecasted changes to temperature and nutrient availability have implications for trophic interactions, and may ultimately influence the access to and palatability of the forage for grazers. Depending on what anthropogenic change will be most pronounced in the future (increase in nutrient deposits, warming, or a combination of them both), different shifts in community dominance hierarchies may occur. Generally, this study supports the productivity–diversity relationship found across arctic habitats, with community diversity peaking in mid-productivity systems and degrading as nutrient availability increases further. This is likely due the increasing competition in plant–plant interactions and the shifting dominance structure with grasses taking over the experimental plots, suggesting that global change could have high costs to biodiversity in the Arctic.

Introduction

Global change is expected to lead to widespread biome and biodiversity shifts across spatial scales, from the regional to the global (Sala, Chapin & Armesto, 2000; Grimm et al., 2013). Many of the fastest changes in physical conditions are predicted to occur in polar and alpine ecosystems, including increasing growing season length, permafrost degradation, and increasing nutrient mobilization, due to a climate warming that is unprecedented in the last two millennia (IPCC, 2007; Kaufman et al., 2009). As a result, these ecosystems are assumed to be particularly vulnerable to climate change (Callaghan & Jonasson, 1995), with some species even going extinct (Klein, Harte & Zhao, 2004). Observational studies have already shown shifts in plant community structure over the last several decades of climate warming in high-latitude and high-elevation tundra, particularly the proliferation of shrubs and grasses (Capers & Stone, 2011; McManus et al., 2012; Callaghan et al., 2013).

Ecosystem responses to global change are complex, nonlinear, and spatially and temporally heterogeneous. Warming is predicted to be the largest driver of change in arctic, alpine, and boreal regions, but nitrogen deposition is also expected to have a large effect, especially in alpine ecosystems (Sala, Chapin & Armesto, 2000). Within a single landscape, warming and nutrient amendment may change in their relative importance from low to high elevations (Graglia et al., 2001). The effects of both have been examined experimentally. Early analyses and meta-analyses of experimental warming in alpine and arctic systems found immediate phenological changes, short-term responses in terms of plant growth, and medium- and long-term responses in terms of plant reproduction and community structure (Arft et al., 1999; Van Wijk et al., 2004; Hollister, Webber & Tweedie, 2005). Nutrient enhancement in these systems also produced short-term growth responses but were sometimes followed by declines in abundance (Dormann & Woodin, 2002; Campioli, Leblans & Michelsen, 2012). There are many potential explanations for the complexity of these responses. The changes themselves create biotic effects such as increased plant competition and changes in litter accumulation, which may in turn affect demography (Foster & Gross, 1998; Olsen & Klanderud, 2014). Species also exhibit different degrees of phenotypic plasticity, and may thus vary in their ability to succeed, survive and thrive under the anticipated changing conditions (Campioli, Leblans & Michelsen, 2012). More recent meta-analyses of temperature manipulation experiments have shown that responses vary sizes to warming treatments and may increase over time, likely due to a combination of all of these factors (Elmendorf et al., 2012). Longevity may also play a role, as short lived species have been predicted to be more sensitive to climate change than more long-lived species (Morris et al., 2008). This implies that many alpine and Arctic plant species could buffer against climate change due to their long-lived nature. In the longer run, however, the long life span of arctic and alpine plants in combination with their capacity for sexual reproduction will determine their fate as evolutionary adaptation is a slow process in comparison with the projected pace of warming (Molau, 1993). It is questionable whether evolution can keep pace with climate change on global scale, thus increasing the extension risk (Jump & Peñuelas, 2005; Parmesan, 2006).

While dividing plant species by functional type may not always yield consistent results within a group (Dormann & Woodin, 2002), the size and speed of responses to simulated global change may nonetheless be somewhat generalizable by plant functional type. For instance, grasses are commonly increasing in abundance under both warming and nutrient treatments (Graglia et al., 2001; Klanderud & Totland, 2005; Campioli, Leblans & Michelsen, 2012). Shrubs have also been detected as expanding in the arctic in recent years (McManus et al., 2012). Another important functional group is cushion plants, which have great influence on ecosystems in polar and high alpine areas throughout the world as they often function as facilitator species across trophic levels (Cavieres & Arroyo, 2002; Molenda, Reid & Lortie, 2012; Roy et al., 2013). Unfortunately, there are very few experimental studies on climate change impact on cushion plants, but the few that exist have shown contrasting responses to warming (Day et al., 2009; Alatalo & Little, 2014). The ability of functional groups to compete for light, nutrients, and other resources varies, and the responses may depend on interactions with co-inhabiting species; for instance, the most abundant (“dominant”) species or functional group in a community can have a strong influences on the biotic conditions of the other species by either negative, competitive interactions or by positive, facilitative interactions (Grime, 1998; Klanderud & Totland, 2004). For instance, an increase in shrub cover may lead to a decrease in species richness (Pajunen, Oksanen & Virtanen, 2011) while the presence of nitrogen-fixing legumes facilitates a richer plant community (Olsen, Sandvik & Totland, 2013). With changes in abiotic conditions, dominant species in more productive alpine plant communities may monopolize added N and P at the expense of their neighbors (Onipchenko et al., 2012), or may show changes in both their competitive response and competitive effect under experimental warming (Niu & Wan, 2008). Thus the redistribution of vegetation types in arctic and alpine ecosystems can create major shifts in dominance hierarchies (Klanderud & Totland, 2005), resulting in feedback loops accelerating changes in ecosystem structure and functioning (Graglia et al., 2001).

There are a growing number of studies on simulated global change effects on alpine plant communities at the community level, by warming and by nutrient addition. However, at present there are only a few factorial studies with experimental warming and nutrient addition on alpine plant communities (for example, Chapin et al., 1995; Alatalo, 1998; Klanderud & Totland, 2005; Jägerbrand et al., 2009; Campioli, Leblans & Michelsen, 2012), and not one of them attempts to assess if short term (<5 years) responses are consistent with medium (6–10 years) or longer term (>10 years) responses. This represents a notable gap in knowledge, as an Alaskan study suggests that short term responses are poor indicators of longer term studies (Hollister, Webber & Tweedie, 2005). We used a factorial experimental design to assess community and functional group response of vascular plants to warming and nutrient perturbations in northern Sweden over a period of seven years. The abundance of lichens, bryophytes, and vascular plants have already been shown to have changed after five years of manipulations in this experiment (Alatalo, 1998; Molau & Alatalo, 1998; Jägerbrand et al., 2009). In this study, we examine whether short-term responses are good predictors for longer-term changes in the system, i.e., are short-term responses consistent with longer-term responses.

Materials & Methods

Study area

Fieldwork took place at the Latnjajaure Field Station (LFS) in northern Sweden, at 1000 m elevation in the valley of Latnjavagge (68°21′N, 18°29′E). Continuous climate data were provided from the early spring of 1992 onwards. Climate is classified as sub-arctic (Polunin, 1951) with snow cover for most of the year, cool summers, and relatively mild, snow-rich winters. Mean annual temperatures ranged from −2.0 to −2.7 °C between 1993 and 1999, with winter minima of −27.3 to −21.7 °C. Mean annual precipitation during this time period was 808 mm, with individual years ranging from a low 605 mm in 1996 up to 990 mm in 1993. The warmest temperatures come in July, which had mean temperatures ranging from +5.4 °C in 1992 to +9.9 °C in 1997.

Physical conditions in the valley vary from dry to wet and poor and acidic to base-rich, with a variety of plant communities to match. This field experiment focused on a meadow community. Previous work in the valley has shown that despite a geographic situation of subarctic-alpine, vegetation of the area is more representative of the Low Arctic, with Cassiope tetragona, Dryas octopetala, and Carex bigelowii among the dominant species (Molau & Alatalo, 1998). At the beginning of this field experiment, the plots were characterized by sedges, shrubs, and cushion plants: C. tetragona, C. bigelowii, Carex vaginata, Silene acaulis and Vaccinium vitis-idaea were present in every plot in the meadow community, while Polygonum vivparum and D. octopetala were present in 75% or more plots.

Experimental design

In July 1995, 20 plots (1 × 1 m) with homogenous vegetation cover were chosen in the meadow plant community and randomly assigned to treatments in a factorial design. There were 8 control (CTR) plots and 4 plots for each of the experimental treatments: warming (T for temperature enhancement), nutrient addition (N) and combined warming and nutrient addition (TN). Warming was induced by Open Top Chambers (OTCs) that increase temperature by 1.5 to 3 °C compared to control plots with ambient temperature (Marion et al., 1997; Molau & Alatalo, 1998). Nutrient addition consisted of 5 g of nitrogen (as NH4NO3) and 5 g of phosphorus (P2O5) per m2, dissolved in 10 L of meltwater. In 1995 all plots were analyzed with a point-frame method (Walker, 1996) to determine the species occurrences under natural conditions before implementing the experimental treatments. The OTCs were then left on plots with warming treatments year-around, and nutrient addition was applied directly after the initial vegetation analyses in 1995 and a few days after snow melt in the subsequent years (1996–2001).

Measurements

All vascular plants in the plots were identified to species level and cover of each species was assessed using a 1 × 1 m frame with 100 grid points (Walker, 1996) in the middle of the 1995, 1999, and 2001 growing seasons. To ensure accuracy and reproducibility, the same grid frame was used for each measurement, and fixed points at the corner of each plot allowed the frame to be placed in the same position within the plot at each different measuring point. Only the first hit of each species was recorded. This method has been shown to be accurate in detecting changes in tundra vegetation (May & Hollister, 2012).

Data analysis

From the point-frame data, we summed the number of touches to pins within each plot to produce plot-level cover measures for each species, which were aggregated into total cover for each plot. Species richness was tallied as the total number of species present at the 100 points within the plot. The cover data, showing the number of hits for each species, were used to calculate the Shannon diversity index and Pielou’s evenness index in each plot (Oksanen et al., 2012).

For each response variable, normality and homogeneity of variance were assessed using standard diagnostic procedures. All statistical analyses were performed in R version 2.15.3 (R Core Team, 2013). A mixed-effects model with fixed factors of nutrient and temperature manipulation, random factors of year and plot was used to analyze responses in total cover, species richness, diversity, and evenness for the whole community using the lme4 package (Bates, Maechler & Bolker, 2012), using restricted maximum likelihood (REML). A generalized linear mixed-effects model using Poisson errors was used for total cover and species richness. Diversity and evenness were normally distributed and a generalized model was not necessary. Backward model selection was performed using second-order AIC (AICc) scores (Mazerolle, 2013) due to the small sample size. For model validation, we examined residuals and q-q plots. Where the interaction of the fixed factors was significant, multiple comparisons were performed within the generalized linear model framework using the glht function of the multcomp package (Hothorn, Bretz & Westfall, 2008).

We also analyzed responses for each of six functional groups: cushion plants, deciduous shrubs, evergreen shrubs, forbs, grasses, and “sedges” (including both Juncaceae and Cyperaceae). We used each species’ pin-hits to calculate each functional group’s cover, and from this its relative cover as a percentage of the total cover in each plot. Functional group cover was analyzed using the same generalized linear model as total cover, described above. Shannon diversity was calculated separately for the deciduous shrub, evergreen shrub, and forb functional groups and analyzed using mixed-effects models as described above. Cushion plant, grass, and sedge functional groups rarely had more than one or two species present in a plot, and as a result analyzing the Shannon diversity lacked utility. Instead, for each plot we used the more simplistic measure of species richness for these three functional groups, categorizing the change from 1995 to 1999 and from 1995 to 2001 as either losing species richness, maintaining the same number of species, or gaining species richness. The distribution of these responses between treatment groups was compared to what would be expected based on cell size and the global mean using Fisher’s exact test, with p-values based on 10,000 replicates of Monte Carlo simulation.

Results

The model selection results for mixed-effects models of all total community and functional group responses are summarized in Table 1. Treatment effects from linear comparisons within the selected model are described below.

Table 1 Stepwise selection of generalized linear mixed-effects models for community responses to simulated global change, including plot and year as random factors.

AICc values for models are listed, beginning with the most complex model (factorial: nutrient × temperature manipulation) and moving backward until the best model is found. This process tests first an additive model (nutrient + temperature manipulation), then univariate models (nutrient manipulation only; temperature manipulation only) and finally a random effects model including only the random factors. The AICc of the best model is highlighted in bold, and the marginal R2 (explaining variation from only the fixed factors) of the best model is also listed.

		AICc values for models in backward stepwise selection	Marginal R2 of
best model	
	Response	Factorial	Additive	Nutrient	Temperature	Random		
Total community	Cover	73.53	76.92				0.45	
	Richness	48.68	45.68	47.21	44.95	45.69	0.07	
	Diversity	39.63	36.91	33.32	32.53	29.17	n.a.	
	Evenness	−111.86	−117.98	−125.69	−125.73	−133.43	n.a.	
Cushion plants	Cover	123.52	125.79				0.41	
Deciduous shrubs	Cover	167.47	165.25	164.49	165.33	165.30	0.11	
	Diversity	72.19	68.59	64.89	74.37	71.81	0.24	
Evergreen shrubs	Cover	200.60	199.21	197.77	205.27	204.55	0.32	
	Diversity	63.43	60.77	57.07	57.08	53.67	n.a.	
Forbs	Cover	160.80	159.16	157.18	159.26	157.09	n.a.	
	Diversity	82.67	80.17	79.06	76.26	75.38	n.a.	
Grasses	Cover	197.65	198.01				0.34	
Sedges	Cover	154.35	152.76	150.74	155.64	154.14	0.15	

Seven years of experimental warming and nutrient addition had a significant interactive effect on total cover of vegetation in the plots. All experimental treatments showed cover differences from the control plots, with the temperature and combined temperature and nutrient treatments decreasing compared to the control plots while the nutrient-only treatment showed a cover increase compared to the control plots (Fig. 1A). A total of 51 species were observed in plots over the course of the seven-year experiment, with individual counts per plot ranging from 6 to 21 species at a given time point. The difference between species richness in warmed and unwarmed plots was only marginal (linear comparisons, p = 0.07; Fig. 1B). This corresponded to no significant effects of any of the treatments on either Shannon diversity (Fig. 1C) or Pielou’s evenness (Fig. 1D).

Figure 1 Total cover, species richness, Shannon’s diversity, and Pielou’s evenness within the meadow community.

Total cover (A), species richness (B), Shannon’s diversity (C), and Pielou’s evenness (D) in the control (CTR), nutrient addition (N), warming (T), and combined nutrient addition and warming (TN) plots in the meadow community. Means are separated by measurement year, with a white bar for 1995, a grey bar for 1999, and a black bar for the final measurement in 2001. Labels for treatments in (A) represent groupings based on significant (p < 0.05) differences from multiple comparisons performed within the generalized linear mixed-effects model. There were no significant differences between treatments for the other response variables. Error bars represent one standard error of the mean within each treatment and year.

Drastic shifts in dominance structure were observed in the nutrient and combined temperature and nutrient manipulation plots over the course of the 7-year experiment (Fig. 2), with grasses increasing in the nutrient and nutrient plus warming treatments, while sedges and deciduous shrubs decreased in cover.

Figure 2 Cover of different functional groups, by treatment and year.

Percentage of the total cover within the plots made up by six different functional groups, by treatment and year.

Cover of cushion plants responded to a significant interaction between the nutrient and temperature manipulations, with the cover in the combined treatment plots significantly lower than in any of the other plots (linear comparisons, p < 0.001; Fig. 3A). In both 1999 and 2001, 20% of plots across the entire experiment had decreased in species richness compared to 1995, whereas the rest had maintained the original number of species (Fig. 5A). No plots gained species of cushion plants. The distribution of the losses between treatment types was not different than that expected by chance (Fisher’s exact test, p > 0.20).

Figure 3 Total cover of cushion plants, deciduous shrubs, evergreen shrubs, forbs, grasses, and sedges.

Total cover of cushion plants (A), deciduous shrubs (B), evergreen shrubs (C), forbs (D), grasses (E), and sedges (F) within the plots. Bar colors and treatment codes are as in Fig. 1. Letter labels above the bars for treatments, where present, indicate that linear comparisons performed within the generalized linear mixed-effects model showed significant (p < 0.05) differences between treatments. There were no significant differences between treatments for the other response variables. Error bars represent one standard error of the mean within each treatment and year.

The effect of nutrient manipulation was included in the best model for cover of both deciduous and evergreen shrubs. For deciduous shrubs, there was no significant difference between cover in plots with and without the nutrient treatment (linear comparisons, p = 0.07, Fig. 3B), however diversity declined significantly in the plots which had added nutrients (linear comparisons, p < 0.001; Fig. 4A). Conversely, evergreen shrub cover decreased significantly with the nutrient manipulation (linear comparisons, p < 0.001, Fig. 3C), but diversity of evergreen shrubs showed no response to any of the treatments (Fig. 4B).

Figure 4 Shannon’s diversity index for deciduous shrubs, evergreen shrubs, and forbs.

Shannon’s diversity index for deciduous shrubs (A), evergreen shrubs (B), and forbs (C) within the plots. Bar colors and treatment codes are as in Fig. 1. Letter labels above the bars for treatments in (A) indicate that linear comparisons performed within the linear mixed-effects model showed significant (p < 0.05) differences between treatments. There were no significant differences between treatments for (B) or (C). Error bars represent one standard error of the mean within each treatment and year.

Forb cover (Table 2, Fig. 3D) and diversity (Fig. 4C) in the plots was unaffected by any of the manipulations.

Grass cover responded to a significant interaction between the nutrient and temperature manipulation. Grass cover increased in the nutrient treatment compared to the control treatment (linear comparisons, p = 0.004), with intermediate abundance in the other plots (Fig. 3E). By 1999, seven of the treatment plots had increased in richness but none of the control plots had changed in richness, which represented a significant effect of the perturbations (Fisher’s exact test, p = 0.002; Fig. 5B). By 2001, additional gain and loss of species richness had negated this effect (Fisher’s exact test, p > 0.05). Sedge cover increased significantly in the plots receiving nutrient amendment (linear comparisons, p = 0.01), especially in 1999 although the effect had waned by 2001 (Fig. 3F). The majority of plots either decreased in species richness or maintained the same number of species by 1999 and 2001, and the distribution of changes among the treatments was not different than that which would be predicted by the global mean (Fisher’s exact test, p > 0.10; Fig. 5C).

Figure 5 Changes in species richness from 1995 levels for the low-diversity functional groups of cushion plants, grasses, and sedges by treatment and year.

Changes in species richness from 1995 levels for the low-diversity functional groups of cushion plants (A), grasses (B), and sedges (C) by treatment and year. Fisher’s exact test showed that for grasses (B), treatment significantly (p = 0.002) affected the gain or loss of species by 1999, but for the other functional groups the gain or loss of species within the treatments was not significantly different than predicted by the global mean.

Discussion

Total vascular plant cover in the alpine meadow increased significantly with nutrient perturbation over the seven-year experiment, maintaining the direction of its short-term response into the medium-term. The most notable responses to simulated global change came at the functional group level, where cover and diversity of some functional groups showed consistent short- and medium-term responses to perturbations (nutrient addition, warming and combined nutrient addition and warming) while after seven years of perturbations others showed either recovery from their initial responses, or intensification of those responses. In particular, the nutrient and the combined warming and nutrient treatment caused changes in the dominance structure in the meadow. Cover of grasses increased dramatically in the nutrient and the combined warming and nutrient enhancement treatments in the meadow community, with response increasing over the course of several years. This increased their relative dominance compared to the previously shown shorter-term responses (Alatalo, 1998; Jägerbrand et al., 2009). These results are in line with other studies, as graminoids have been reported to increase dramatically in abundance in response to nutrient addition in several previous studies in alpine and arctic communities (Theodose & Bowman, 1997; Klanderud & Totland, 2005; Calvo et al., 2005; Campioli, Leblans & Michelsen, 2012; Onipchenko et al., 2012). Sedges that traditionally have been incorporated into the “graminoids” functional group in many previous studies showed a contrasting pattern, with abundance decreasing among years in all treatments in the meadow community. This is in contrast to other studies that have indicated that sedges may have more positive responses than grasses (Bowman et al., 1993; Walker et al., 2001; Soudzilovskaia & Onipchenko, 2005; Bassin et al., 2007). These studies have suggested that the positive response is because traits such as lower nutrient losses and slow turnover rates are more important in nutrient limited habitats for competitive success (Aerts, 1999). Furthermore, it has previously been reported that species respond differently to temperature and nutrient perturbations at different sites (Elmendorf et al., 2012; Press et al., 1998), thus the species composition of the “functional group” at a specific site may influence the community’s responses. Indeed, the functional group designation has not always yielded consistent results in global change experiments (Dormann & Woodin, 2002). In that case, a possible explanation for our contrasting results may be that the sedge species found in our meadow community might not be as responsive as the sedge species from other sites reporting a positive response for the functional group.

Previous short-term studies have found positive short-term responses of forbs to nutrient addition (Henry, Freedman & Svoboda, 1986; Bowman et al., 1993; Calvo et al., 2005; Onipchenko et al., 2012), including a five-year study in this same community (Jägerbrand et al., 2009). However, we found that this response had disappeared after seven years of perturbations. In all treatments, mean forb cover decreased to a level near or below its initial starting value. Warming also caused contrasting short- and longer-term responses: after seven years of warming the forbs had declined their cover, while having previously not responded to shorter-term treatment (Jägerbrand et al., 2009). Contrasting responses were also found in a short-term study in the Swiss Alps, where species-specific responses of different forbs to nutrient addition varied between negative, neutral and positive (Bassin et al., 2007).

Evergreen shrubs showed a significant and complex response to nutrient addition. After seven years the cover of evergreen shrubs had recovered from the short-term negative response to the combined warming and nutrient addition that was reported in a previous study (Jägerbrand et al., 2009), gaining their previous relative share of the dominance hierarchy in terms of cover. However, cover had increased in the control and temperature treatments over seven years, an effect which seemed to be dampened by the nutrient perturbation. Nevertheless, the appearance of a recover by evergreen shrubs is interesting as, for instance, in a four-year study in Norway the evergreen shrub Dryas octopetala lost its dominant position in the community to graminoids in response to nutrient addition and combined warming and nutrient addition (Klanderud & Totland, 2005). It has been suggested that evergreen shrubs are more likely to decline in response to nutrient addition, while deciduous shrubs are likely to increase due to the same perturbation (Chapin et al., 1995). The potential recovery of evergreen shrubs in our results is a novel finding, and should be further examined in other long-term studies. Furthermore, we found no support for a deciduous shrub increase. Rather, deciduous shrubs cover decreased in response to both the nutrient and combined warming and nutrient addition treatments. This was caused by an initial short-term response (Jägerbrand et al., 2009), since their relative share of the cover did not continue to decline after the five years. These results reinforce previous experimental findings that diversity of both types of shrubs are negatively affected by increasing nutrient availability (Press et al., 1998; Klanderud & Totland, 2005).

Cushion plants decreased in cover in response to nutrient and the combined warming and nutrient addition. Similarly, in high Arctic Svalbard, 5 years of nutrient addition caused significant decrease of Saxifraga oppositifolia (Robinson et al., 1998), while Silene acaulis has been shown to respond in contrasting manner to short and medium term nutrient addition (Alatalo & Little, 2014). If cushion plants begin to decrease in larger numbers in severe environments, this could potentially impact a wide array of species in ecosystems where they are found due to their importance as facilitator species (Cavieres & Arroyo, 2002; Molenda, Reid & Lortie, 2012).

Total species richness declined over the seven years of warming, while species richness, diversity, and evenness showed nonsignificant decreases in the combined nutrient and warming treatment. The largest decline in species diversity after seven years of perturbation was found in deciduous shrubs in response to nutrient addition and the combined warming and nutrient addition. In contrast grasses increased their species richness, almost tripling in response to the combined warming and nutrient addition, and sedges showed a nonsignificant trend of increasing species richness in response to the nutrient addition but decreasing in response to warming. A decrease in species richness due to simulated global change has also been reported in other studies. A 9-year study with experimental warming and nutrient addition in Alaskan tundra found that species richness declined by 30–50% due to losses primarily of rarer species, but this was mainly caused by loss of bryophytes, lichens and forbs (Chapin et al., 1995). In alpine Norway, four years of combined warming and nutrient addition caused a significant decline in total species richness, caused by a decline in bryophytes and lichens, while the same perturbation increased species richness of graminoids (Klanderud & Totland, 2005). In the same study species richness of forbs increased in response to nutrient addition. The contrasting results of species richness of forbs ranging from negative (Chapin et al., 1995), neutral (this study), to positive (Klanderud & Totland, 2005), suggest that the responses may be highly species-specific.

Community diversity has been shown to decrease in arctic and alpine meadows in response to nutrient addition (Theodose & Bowman, 1997; Wardle et al., 2013) and in particular in response to combined warming and fertilization (Press et al., 1998; Klanderud & Totland, 2005). Generally, this study supports the productivity–diversity relationship found across arctic habitats, with community diversity peaking in mid-productivity systems and crashing as nutrient availability increases further (Virtanen et al., 2013). This is likely due to the increasing competition in plant–plant interactions and the shifting dominance structure with grasses taking over the experimental plots and suggests that global change in the arctic could entail not only redistribution of vegetation types, but also significant costs to biodiversity.

Conclusions

The different perturbations caused shifts in dominance hierarchies in the alpine meadow. Nutrient addition drove the community to become more dominated by grasses, sedges and forbs. Short-term responses were shown to be inconsistent in their ability to predict medium-term responses for sedges, shrubs, cushion plants and forbs. However, grasses showed consistent and very substantial response to nutrient addition over the whole period of seven years. The non-linear responses over time point out the importance of longer-term studies with repeated measurements to be able to better predict future changes. The non-linear responses also have important implications for improving modeling the future changes to global change. The different changes to warming and nutrient addition will likely have implications for trophic interactions, and may ultimately influence the access to and palatability of the forage for grazers. Depending on what anthropogenic change will be most pronounced in the future (increase in nutrient deposits, warming, or a combination of them both), different shifts in community dominance hierarchies may occur.

Supplemental Information

Supplemental Information 1 Raw data

Raw data used for the analysis.

Click here for additional data file.

The authors thank the staff of Abisko Scientific Research for their help and hospitality, and Vivian and Björn Aldén for assistance in the field. The authors thank Hannah Buckley and two anonymous reviewers for constructive comments that improved the manuscript.

Additional Information and Declarations

Competing Interests

Author Contributions

Field Study Permissions

The authors declare that they have no competing interests. Annika K. Jägerbrand is an employee of the VTI, Swedish National Road and Transport Research Institute.

Juha M. Alatalo conceived and designed the experiments, performed the experiments, wrote the paper, reviewed drafts of the paper.

Chelsea J. Little analyzed the data, wrote the paper, prepared figures and/or tables, reviewed drafts of the paper.

Annika K. Jägerbrand performed the experiments, reviewed drafts of the paper.

Ulf Molau conceived and designed the experiments, performed the experiments, reviewed drafts of the paper.

The following information was supplied relating to field study approvals (i.e., approving body and any reference numbers):

Our research did not require a permit.

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
