# Peer review of "Dominance hierarchies, diversity and species richness of vascular plants in an alpine meadow: contrasting short and medium term responses to simulated global change"

_PeerJ, doi:10.7717/peerj.406_

## Round 0.1 · original submission · Major Revisions

Please review your data analysis methods in the light of my comments below and those of the second reviewer. Either you need to change the way you have modeled the data so that you deal correctly with the nested design, or, if you did in fact do an appropriate mixed model analysis, you need to better explain what you did so it is clear to readers. My more minor comments on the manuscript are also given below.

Line 19: If the community became dominated by sedges, then it seems incongruous to say that sedges decreased in the next sentence. Did they increase or decrease? If the increase was relative but their absolute change was to decrease, then this needs to be clearer. In the results you say that sedge richness increased in the nutrient-only treatment.

Line 28: “drastic costs” seems like awkward wording. Can you use “high costs” or something similar?

From Line 93 (suggested rewording for grammar): “…and not one of them attempts to assess if short term responses are….Alaskan study suggests…”

Line 106 (suggested rewording): “…climate data were provided…”

Line 110: “…winter minima…”

Line 114: delete extra tab

Line 116: “…work in the valley has shown that…”

Line 124 (extra space needed): “…plots (1 x 1 m)…”

Line 138: What was the size of the grid? I assume that you had a grid within the 1x1m plot? This part is not clear.

Line 139: “…grid frame was used…”

Lines 150-158: Your description of the statistical analysis needs more work because there are several aspects that are not clear:
1. What transformations were applied and to what variables? None of your results graphs are presented using transformed variables, so the reader cannot figure out what was done.
2. Why did you need to use lme4? Did you do a mixed-effects model (with a random effect) or were all your variables fixed effects? If you just needed to perform a generalized linear model then you could have used the function ‘glm’.
3. You say above that you calculated relative cover, but you don’t describe absolute cover anywhere. What do you mean here exactly? Why analyse cover data using Poisson errors (and I assume a log link)?
4. Why switch to ANOVA for richness? Because richness is count data, it is more correctly modelled using Poisson errors.
5. It is no longer recommended to use likelihood ratio tests with generalized linear models to assess significance. In your case, it would be appropriate, given the small sample size to use the small sample size corrected version of AIC: AICc (see David Anderson’s 2008 book). You can incorporate contrasts into this modelling (rather than using post-hoc testing), which is appropriate given that you have a fully replicated experiment.
6. Line 162: your symbol for chi-square has not worked. Try using ‘normal font’.
7. Should Figure 1 A and Figure 2 A and B have error bars?
8. What values do the error bars on Figure 3 represent?
9. Line 211: This first sentence of the discussion is somewhat opaque. You could give some more detail here to refresh readers on what your aims were and to accommodate those who haven’t read your results in great detail.
10. Line 233: This sentence would read better if it was reworded so that there aren’t so many repeated words.
11. Line 257: Reword the sentence “…shrubs are tend to increase…”
12. Line 283: “…Norway, four…”
13. Line 304: “…is likely due to the…”
14. Line 308: “The different perturbations caused…”

Reviewer 1 ·

Basic reporting

This manuscript by Alatalo et al have studied the effect of warming and nutrient-addition in an arctic meadow over a seven years period using an experimental set-up. They conclude that adding nutrients, and nutrients+warming, result in drastic shifts in dominance hierarchies and that these shifts go in different directions. The point out that these shifts will affect herbivory-interactions (more grasses, more grazing), and facilitation of other species (less cushion-plants) under future climate scenarios.
Further, they show that there are large seasonal variations, which underscores the importance of long-term studies that include repeated measurements to predict future changes. I actually think this latter finding is most interesting, and deserves to be highlighted more in this manuscript.
The manuscript was well written, and easy to read, and have several interesting findings that deserve to be published. I have some general (and some more specific) comments/suggestions, which I think can improve the manuscript further before publishing.
The abstract is nice and clear, but I miss some information about the amount of data the conclusions are based upon (number of plots, times visited…) and a clearer difference between trends and significant findings, so the reader fast and easy can evaluate the strength of the conclusion. I also find it puzzling that the conclusion in the end of the paper highlights other findings than what is highlighted in the abstract (Conclusion: variation among years, and increased grasses and the effect on herbivores, abstract: reduction of cushion plants/facilitation).
Introduction
The introduction is overall very good. I miss one thing though, and that is an introduction to the effect of functional groups like cushion plants and grasses, as they end up as discussion points.
(line 93/94) I also think to use the “short-term” about a five year study, and “long-term” about a seven year study is rather arbitrary and misleading. I suggest using the actual number of years when referring to other multi-year studies, or at least define what a short-, medium- and longer-term study is.
Material and methods/Results
The number of replicates per treatment is very low (four treatment/eight control), and much lower than what is suggested e.g. by the itex manual that they refer to for the vegetation analyses. This issue should be addressed in some way.
line 149. The authors state that they use species richness data to calculate Shannon diversity. They do have cover-data as well, but this is not presented in the results, which seems strange. The Shannon could be calculated based on the cover (abundance) data.
For the figures, I find it a bit difficult to understand which grouping the letters and numbers refer to, eg in figure 3.
Arctic or Alpine used as an adjective, like in 219 and 283, should be arctic and alpine
line 281 caused instead of cause

Experimental design

The number of replicates per treatment is very low (four treatment/eight control), and much lower than what is suggested e.g. by the itex manual that they refer to for the vegetation analyses. This issue should be addressed in some way.

Validity of the findings

See comments given under basic reporting

Additional comments

See comments given under basic reporting

Reviewer 2 ·

Basic reporting

The study meets most of the criteria.

The figures need the following improvements:
-All figures need to be redrawn to fit one column. You should resize the label font to make the axes readable.
-The lines in the figures should be changed to different formats to allow b/w printing.
-All figures reporting averages should also show a measure of variability (SE, SD or CI).
-I would split figure 1 into two separate figures.

Experimental design

-Please provide more details on the method to estimate the species cover.

-The main statistical analyses (GLM) seem to be not correct. The design included repeated measures (three times in each plot). However, the model employed did not include any random factor. The main effect of the treatments should be tested using n= number of plots and not n=number of observations. You should run a general linear mixed model with plot ID as random factor. Please also indicate the estimation method (ML or REML?).

Validity of the findings

See comment above on the analyses.
No further comments.

Additional comments

The study is generally well-written and presents a nice data-set. I have only a few comments to improve the manuscript. Although the level of replication of this study is quite low the lack of long-term studies on the topic makes this contribution interesting. Along with the comments reported I have a few suggestions:
- I would like to see more discussion on the non-linear response of several functional groups. It is quite interesting that some species groups first increased and then declined within the same treatment.
-I would also suggest to add a real evenness analysis to test how dominance changes with your treatment.
-If species composition varies enough it would be also interesting to investigate variation in species replacement over time within each treatment. There are several indexes based on both presence/absence and abundance data.

---

## Round 0.2 · Minor Revisions

Thanks for your comprehensive response to the reviewer comments. I just have a few more minor changes that will help improve things further:

Line 120: "The OTCs were..." (not OTC's)
Line 139: "Mixed-effects models with fixed...Bolker 2012), with restricted...".
Line 142: "A generalized linear mixed-effects model using Poisson...and species richness. Diversity and evenness..."
Line 147: "general linear hypotheses" This doesn't make sense to me. I assume you mean multiple comparisons? You should describe the procedure and any functions or packages you used to do this part of the analysis.
Figure: Please give descriptions of the treatment letter codes in the captions of your figures. Also, make sure that the colours of the bars are labelled (e.g. Figure 1 we don't know what the three bars are for each treatment).
Lines 162-177: Please describe the direction of the treatment effects where possible. e.g. the warming treatment had a negative effect on species richness.

---

## Round 0.3 · accepted · Accept

Thanks for your fast response to those last changes. I think that the paper looks great!